# A protocol to simultaneously examine cardiorespiratory, cerebrovascular and neurophysiological responses inside a hypobaric chamber

Evan A. Hutcheon[1]*, Sherri Ferguson[2], Victoria E. Claydon[1], Urs Ribary[3‡], Sam M. Doesburg[1,4‡]

1 Department of Biomedical Physiology and Kinesiology, Simon Fraser University, Burnaby, British Columbia, Canada, 2 Environmental Physiology and Medicine Unit, Faculty of Science, Simon Fraser University, Burnaby, British Columbia, Canada, 3 Department of Psychology, Simon Fraser University, Burnaby, British Columbia, Canada, 4 Institute for Neuroscience and Neurotechnology, Simon Fraser University, Burnaby, British Columbia, Canada

☯ These authors contributed equally to this work.
‡ UR and SMD also contributed equally to this work.
* eahutcheon@gmail.com

**Data Availability Statement:** Due to legal and ethical restriction, data cannot be made publicly available. Additional published or public analyses

## Abstract

We describe a protocol to examine neurophysiological (electroencephalography, EEG), cerebrovascular (ultrasound assessments of middle cerebral artery blood velocity, MCAv) and cardiorespiratory (blood pressure, oxygen saturation, end-tidal gases, respiratory rate) responses inside a hypobaric chamber. This procedure aims to standardize the methodology in experiments conducted within a hypobaric chamber such as comparing normobaric and hypobaric hypoxia. This is important because current understanding of relationships between neurophysiological activity, and cerebrovascular and cardiorespiratory responses under varying environmental conditions remains limited. This procedure combines simultaneous neurophysiological, cardiorespiratory and cerebrovascular evaluations, allowing a comprehensive understanding of electro-neurophysiological activity. Our protocol requires an hour and a half of equipment setup, 1–1.5 hours of participant set-up, and 30 minutes of experimental data collection. Since multiple simultaneous physiological recordings, including EEG in this environment, can be fraught with pitfalls, we also provide practical considerations for experimental design and recording setup. Advanced knowledge of hypobaric chamber operation is required, alongside expertise in EEG and transcranial Doppler ultrasonography. Following our procedure one will acquire simultaneous recordings of neurophysiological, cerebrovascular and cardiorespiratory data.

## Introduction

Hypoxia research is critical to understand the effects on the human body of ascent to high altitude, permanent high altitude living, airline travel and other occupations with a risk for

would only be permitted with ethics approval for secondary data access, which we do not have. When ethics was originally granted for this study we did not state that the data would be used for secondary use. We only stated that the data would be used for a thesis and scientific reports. You can contact our ethics board at dore@sfu.ca regarding our study (#20160635).

**Funding:** The author(s) received no specific funding for this work.

**Competing interests:** The authors have declared that no competing interests exist.

hypoxia, and hypoxic disease at sea level. These investigations are vital because of the widespread effects of hypoxia on the body, with profound impacts on cardiovascular, cerebrovascular, respiratory, neurophysiological, and cognitive responses that can have life threatening consequences.

Hypoxia research is often focused on field studies performed at high altitude, but these tend to be technically and logistically challenging. Alternatives to field studies can involve assessments performed within a hypobaric chamber (hypobaric hypoxia [HH]) or with a hypoxic gas mixture (normbaric hypoxia [NH]). Collecting data inside a hypobaric chamber (Fig 1) is more challenging and has greater risks than utilizing a hypoxic gas mixture. This protocol has broader application for integration of EEG into hypoxia studies using a hypobaric chamber, but is here described as a protocol for comparing HH and NH as this was the purpose for which it was originally developed. NH induces hypoxia by lowering the fraction of oxygen with no change in absolute pressure, and HH induces hypoxia due to a decrease in absolute pressure without the change of fraction of oxygen (i.e., being at a high altitude). Unfortunately, the literature is divided on the impacts of differences between these forms of hypoxia due to methodological differences and potential experimental confound. Data are especially scarce regarding potential neurophysiological differences between NH and HH. A standardization of study methods is needed, especially in terms of recording neurophysiological responses, e.g. using electroencephalography (EEG) alongside other physiological measures inside a hypobaric chamber.

NH is a safer and less expensive option to study hypoxia when compared to HH, and accordingly, understanding whether these two protocols are equivalent has important implications for decisions as to whether hypoxia research can be performed with NH instead of HH. HH requires a hypobaric chamber or field evaluations at altitude; thus HH is more expensive and dangerous, as it carries a risk for barotrauma, acute mountain sickness, and potentially even decompression sickness. NH on the other hand is safer and less expensive, as NH only requires a properly fitted mask and a hypoxic gas mixture to induce hypoxia. Potential physiological or neurophysiological differences between the two forms of hypoxia remain an active area or research[1–5]. Currently, it is assumed that the partial pressure of oxygen is the only

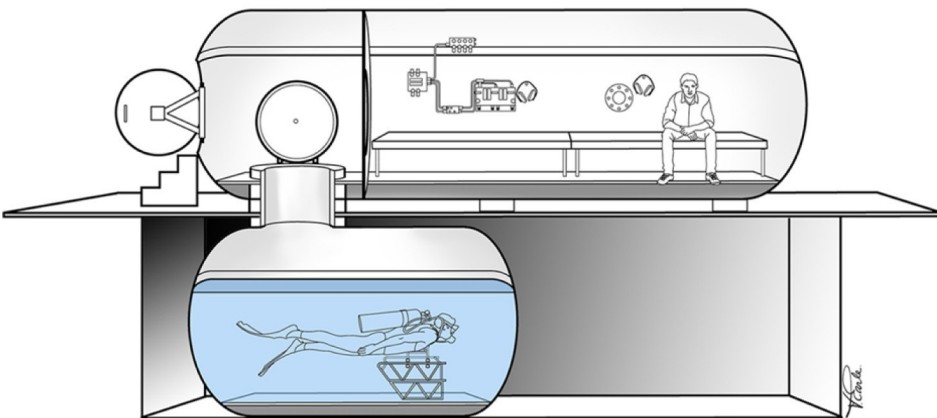

**Fig 1. A schematic of the hypo/hyperbaric chamber in which we performed our three exposures.** The bottom part of the chamber can be utilized for diving studies; however, for our study that section of the chamber was closed. All recordings took place in the main lock of the hypobaric chamber, where the figure is sitting. Adapted from previous publications [1, 2].

relevant factor when considering a NH or HH stimulus [6] resulting in NH studies utilized as surrogates for HH. However, it may be that other factors, such as barometric pressure, can influence physiological responses to HH and NH [4, 6–14]. Previous studies have compared NH and HH; however, interpretation of these investigations is complicated by the lack of standardisation of study methods. This was highlighted by a review analyzing NH and HH cross-over studies in which it was not possible to perform statistical analysis on the data due to study heterogeneity, highlighting the need for method standardisation [11]. In this review, 225 unique articles were identified, highlighting the importance of the work in the field, and yet only 13 of these studies met the defined inclusion and exclusion criteria. Among these 13 studies, only 9 randomized the order of hypoxic exposure and none examined neurophysiological responses to the induced hypoxia. This review highlights the issues with designing a study comparing NH with HH, the need for standardisation of study methodology, and the lack of consideration of neurophysiological outcomes in hypoxia research.

We will describe how we carefully compared NH and HH hypoxia (with use of a nomobaric normoxia [NN] control condition for comparison), providing a protocol for research groups aiming to perform similar research. We combine whole-head EEG with cerebrovascular (ultra-sound assessments of middle cerebral artery blood velocity, MCAv) and cardiorespiratory (blood pressure, oxygen saturation, end-tidal gases, respiratory rate) responses (Fig 2) inside a hypobaric chamber and provide practical advice and troubleshooting advice to maximize the effectiveness of multimodal data collection in this challenging environment.

## Development of protocol

We developed a novel approach to record EEG activity and other physiological measures inside a hypobaric chamber to compare NH and HH. We combined traditional EEG procedures including those using whole-head EEG during cognitive task performance (Fig 2)–in our case visuospatial attention orienting [1, 15–18] with measures of responses to environmental stimuli (i.e., high altitude physiology) to create a procedure that allows us to directly

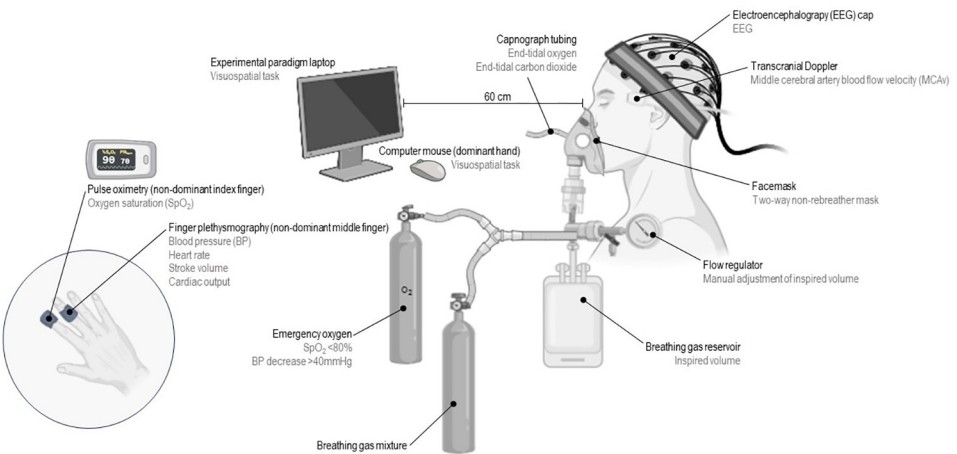

**Fig 2. A schematic showing our experimental set-up.** The participants sat upright with their eyes 60 cm from the computer screen. They had a 64 channel EEG cap on, with a transcranial Doppler ultrasound probe over the transtemporal window and tucked underneath the cap. The participant wore a mask with a modified non-rebreather valve that allowed for side stream sampling of end-tidal gases. The participants left index finger had an $SpO_2$ monitor, their left middle finger had a finger cuff for collecting blood pressure data, and their right hand held the mouse for the task. The inside chamber safety attendant controlled the flow regulator making sure that the breathing gas reservoir bag never collapsed or became too full. Created with BioRender.com.

compare NH to HH. Prior to initiating this line of research we performed a literature review that identified other studies that attempted to compare NH and HH stimuli [19–23], but none had done so with an EEG, and many had methodological confounds.

We selected an equivalent altitude exposure of 3962m (464 mmHg) as this produces a reduction in arterial oxygen saturation (SpO$_2$) to approximately 80% [24–26], which coincides with the plateau in the oxygen dissociation curve [27], producing a robust desaturation that is safe for the participant. Of note, this level of desaturation is also similar to that experienced by individuals with sleep apnea and obesity hypoventilation syndrome [28]. Consultation with a physician with experience in hypobaric medicine revealed additional benefits to this level of hypoxic exposure, as they felt confident that this "mild" level of hypoxia was compatible with having safety provision provided by a paramedic or dive medic technician present, rather than requiring supervision by a physician. This altitude also has practical importance in the context of aviation medicine, as the Canadian Aviation Regulations state that an unpressurized aircraft can be flown without supplementary oxygen for up to 30 min between 10,000–13,000 ft (3048 m-3962 m) [29].

We performed extensive pilot testing to streamline our procedure. We first collected EEG data inside the chamber with no change in pressure to examine the recording quality and evaluate whether the signal-to-noise ratio would be suitable inside the chamber. After confirming this was the case, we confirmed that the hypoxic gas mixture proposed for the NH protocol would indeed produce a decrease in SpO$_2$ to approximately 80%. We then tested the entire protocol during NN conditions inside the chamber to consider technical, recording and space constraints. During these pilot tests one participant experienced a vasovagal reaction, as sometimes occurs during medical-like procedures [30]. Following discussion with a physician specializing in high altitude medicine, we modified our protocol to ask participants to comply with strategies known to diminish the likelihood of a vasovagal reaction, including drinking a large bolus of cold water (500 ml) before the experiment [31–33], and moving their legs during breaks in the protocol to prevent blood pooling in the lower limbs [34, 35]. We also incorporated measurement of the participants' beat-to-beat blood pressure, with continuous real-time monitoring, and additional criteria to stop the test and increase the inspired oxygen fraction to 100% if the systolic blood pressure decreased by >40 mmHg (relative to their baseline recording) to prevent syncopal reactions. To ensure accurate blood pressure and SpO$_2$ values from devices applied to the fingertips we placed a small blanket around the participant, over the arm and hand, to ensure their hands remained warm and circulation to the fingertip was not compromised due to cutaneous thermoregulatory vasoconstriction.

## Applications of method

We have applied this protocol to compare neurophysiological responses between NH and HH in an eyes-open resting state paradigm [2] and during a visuospatial attention task [1]. Other tasks could readily be utilized to address different research questions. As many of the symptoms of hypoxia are cognitive in nature [3, 13, 36, 37] this procedure represents an unparalleled opportunity to examine potential cognitive differences between HH and NH.

Our research question involved comparisons of responses to NH and HH, but this procedure can also be utilized for studies examining responses to hypoxic interventions in general, rather than necessarily to compare NH and HH. Diverse paradigms are possible with this approach, such as investigating hypobaric normoxia which could be utilized to model oxygen breathing at high altitude or during aircraft depressurization with supplementary oxygen. Our paradigm can also be utilized to study acclimatization, acute mountain sickness, and other high altitude specific pathologies. An acclimatization study would involve the participant

having to stay in the chamber for a few days and would require adaptations to the existing protocol such as a commode toilet and food deliveries through the hatch alongside a larger research team that could work round the clock. Furthermore, it can also be used to study other populations of interest, evaluate sex differences in responses to hypoxia, or examine the impact of hypoxia in individuals with health conditions. Outside of the realm of hypoxia research, our experimental set-up can also be utilized for studies investigating the relationship between EEG and cardiovascular/respiratory measures, such as the relationship between respiration and neuronal oscillations across large-scale brain networks [38, 39]; in this case, the experimental setup would be the same as our set-up during the NN component of our study.

## Comparison with other models

The protocol described here builds on previous methods. The primary novelty of our approach relates to utilizing an EEG inside a hypobaric chamber, with concurrent cardiorespiratory and cerebrovascular monitoring, and while directly comparing NH with HH. Previous studies investigating the effects of hypoxic on EEG parameters have done so in HH conditions [40–46] or NH conditions [25, 47–53]. Ours is the first study to compare responses to NH and HH while considering appropriate blinding, controls, and randomisation, and to incorporate EEG responses alongside cardiovascular and cerebrovascular measures [2].

Some prior studies have utilized a hypobaric chamber for HH, while others perform the HH in the field [4]. HH field studies (i.e., on a mountain versus inside a hypobaric chamber) have a range of practical issues including: travel time to altitude, the need for portable equipment, potentially needing a generator to power the gear, and being at the "mercy" of the weather, which in mountainous conditions can change rapidly. Studies performed in a hypobaric chamber allow for control of environmental factors such as humidity and temperature, alongside having a quick "travel" time to altitude (i.e., depressurization) and the general comforts of collecting data inside a laboratory. The major downside of utilizing a hypobaric chamber is that these are specialised facilities that are not widely available except at specialised research centres and hospitals, and the potential cost of hiring a hypobaric chamber can be prohibitive. If NH and HH paradigms were ultimately shown to be equivalent, this would pave the way for further hypoxia research to be performed using the simpler NH paradigm as a more accessible method of hypoxia induction.

## Experimental design

The protocol described here was utilized to collect EEG and cardiovascular and respiratory data to compare NH and HH, including a control NN condition for comparison. This exact protocol requires specific equipment and specific knowledge, though we expect findings to be applicable to a range of comparable designs. We assume that team members will have advanced knowledge of high-altitude physiology and the use of an EEG in cognitive neuroscience contexts. In terms of equipment, access to a hypobaric chamber, EEG, and cardiovascular and respiratory recording equipment are required. It is also necessary to be able to mix the NH hypoxic gas mixtures, or be able to order them. Here we describe the procedure we used, with suggestions for how to improve or adapt it to address different research questions. As hypoxia research carries a certain degree of risk, participant safety is of paramount concern and overrides any research goals. Participants need to be informed regarding potential dangers due to hypoxia in order to provide informed consent to the study, and they need to be aware that they can discontinue the experiment at any time. Given individual differences in hypoxia symptoms, the participant experience in the study can vary. This protocol will provide a framework for conducting studies comparing NH and HH with an EEG to help standardisation of methods.

## Expertise needed to implement protocol

This protocol requires a team with specialized knowledge. Required team members include a clinical or occupational chamber operator, and an assistant chamber operator with the relevant certifications to properly and safely operate the chamber. An experimenter inside the chamber who is trained as a hypobaric chamber safety attendant is needed to administer the testing protocol, ensure signal acquisition for the monitoring equipment, data recording, and to monitor participant safety. The hypobaric chamber safety attendant should be versed in recognizing the signs of pre-syncope and at a minimum have their CPR-C/AED certificate (Emergency Oxygen Provider certification is also recommended). In addition to data collection and participant safety monitoring, the hypobaric chamber safety attendant will ensure compliance with participant safety cut off criteria, ensuring the $SpO_2$ and blood pressure do not fall below the cutoff values, while also controlling the gas flow rate. The hypobaric chamber safety attendant must be trained in the proper use of an EEG, the use of a transcranial Doppler ultrasound, $SpO_2$ measurements, beat-to-beat blood pressure measurements using finger plethysmography, and capnography for end-tidal gas measurements. It is paramount that the hypobaric chamber safety attendant is experienced in using all the recording instruments as they will have to troubleshoot any potential technical issues independently–once the hypobaric chamber is in operation it is not possible for additional personnel to enter or leave the testing space.

A trained medical professional is also needed for participant safety. We consulted with a physician experienced in high altitude medicine who was confident that a dive medic technician or paramedic were adequate for our study, as our hypoxic dose only reduced the participant's $SpO_2$ to approximately 80%. Note that the need for and qualifications of supervising medical personnel depends on the level of the hypoxic stimulus, at a more severe hypoxic dose we recommend having a physician present. As this is a research protocol where there may be unexpected events, it is strongly recommended that the physician with expertise in emergency medicine be present at all times.

## Limitations and practical considerations

For safety, and to ensure participants are effectively able to equalize their ear pressures (to prevent tympanic membrane rupture) it was necessary to do a brief depressurization (to 630 mmHg [1554 m] for approximately 3.5 minutes) during the HH condition (Fig 3). While necessary, this was a limitation to our study, because our participants briefly became hypoxic during the depressurization period. We expected such a short exposure to be insufficient to cause significant desaturation, but in fact participant $SpO_2$ levels dropped to between 80–85% during this time [54]. In future experiments, this could be avoided by having participants breathe 100% oxygen during the brief safety decompression in the HH condition. If this approach were adopted, it would need to also be employed during the NN and NH conditions to ensure a similar stimulus in all three conditions. Alternatively, one could employ the same protocol that we did and simply have participants in the NH condition start breathing the hypoxic gas mixture for 3.5 minutes (same time period as the decompression) before the start of the task so that both conditions start with the participants being hypoxic.

It was necessary for participants to wear a face mask for controlled gas delivery during the NH condition. Accordingly, we required participants to wear the same mask in every condition to ensure a similar physiological stimulus, and maintain blinding. If possible, we recommend future groups fill the chamber with enough nitrogen during the NH condition to reduce the fractional oxygen content to the desired level (in our case 12.8% oxygen), avoiding the need for a face mask for gas delivery. The benefit of this approach is improved comfort for the participant, and better-quality EEG recording, as the straps of the mask may add pressure on

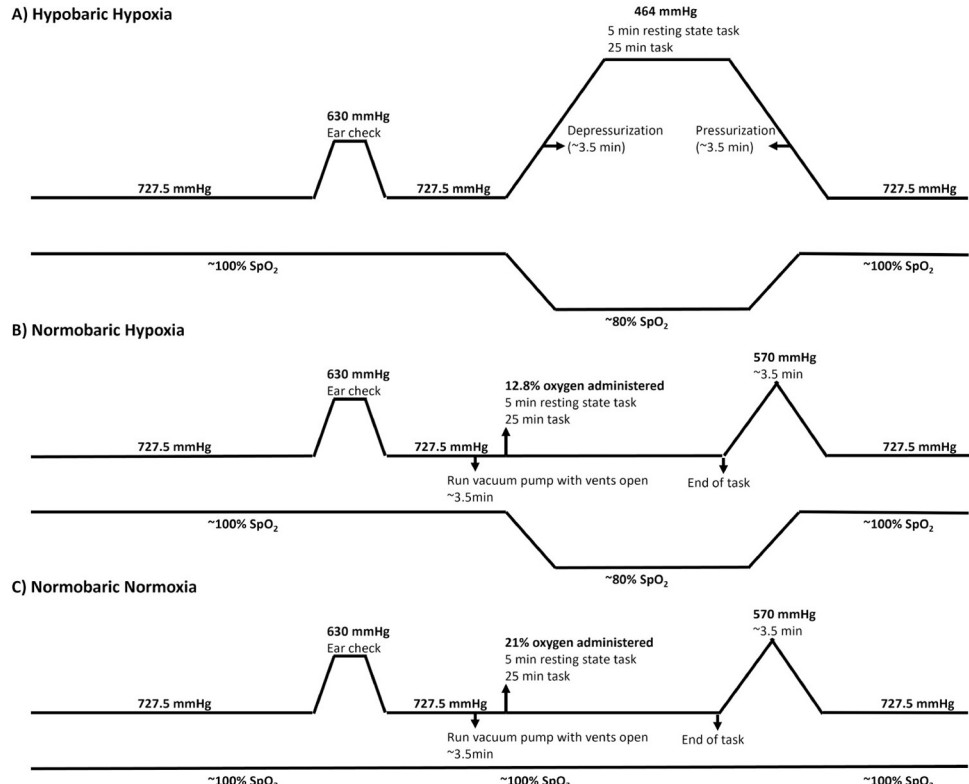

**Fig 3. Comparison of the three conditions.** A) During the hypobaric hypoxia condition participants first performed a depressurization to 630 mmHg (1554 m) to be certain that they could equalize their ears. Following this we then depressurized the chamber to 464 mmHg (3970 m) and had the participants complete the 5 minutes resting state and 25 minute attention task. Following completion, the chamber was pressurized to 727.5 mmHg (366 m). B) During the normobaric hypoxia condition participants first performed a depressurization to 630 mmHg (1554 m) to be certain that they could equalize their ears. Following this we ran the vacuum pump in the hypobaric chamber for 3.5 minutes with the vents open to simulate the noise of depressurization experienced in the HH conditions. After 3.5 minutes we administer the 12.8% oxygen gas mixture to the participants, and they began the 5 minutes resting state and 25 minute attention task. Following completion of the task, the hypobaric chamber was depressurized to 570 mmHg (2360 m) and then pressurized to 727.5 mmHg (366 m) in 3.5 min so that the participants felt a pressure change. C) During the normobaric normoxia condition participants first performed a depressurization to 630 mmHg (1554 m) to be certain that they could equalize their ears. Following this ran the vacuum pump in the hypobaric chamber for 3.5 minutes with the vents open to simulate the noise of depressurization experienced in the HH conditions. After 3.5 minutes we administer the 12.8% oxygen gas mixture to the participants, and they began the 5 minutes resting state and 25 minute attention task. Following completion of the task, the hypobaric chamber was depressurized to 570 mmHg (2360 m) and then pressurized to 727.5 mmHg (366 m) in 3.5 minutes so that the participants felt a pressure change.

electrodes. If no masks are utilized, then a nasal cannula will be needed to record the end-tidal gas levels. The main limitation of the approach is the cost of the nitrogen gas required to manipulate the fractional oxygen content in the entire chamber.

We also manually controlled the air flow to the participant to match their breathing rate. An automated airflow device is recommended instead if possible. Having an automated dynamic end tidal forcing machine would be beneficial, as it would allow investigation of the effects of hypoxia with and without the accompanying hypocapnia. With the experimental protocol described here, we are investigating both hypoxia and the associated hypocapnia secondary to increased ventilation, as we are not forcing the end tidal gas concentrations. While this approach recapitulates the real-life scenario of hypoxic exposure, it does not permit mechanistic consideration of the effects of hypoxia and hypocapnia on the responses obtained.

The transcranial Doppler ultrasound probe was anchored in place using the EEG cap because our usual fixation system was not compatible with the simultaneous recording of EEG. We found that over time re-adjusting the probe was often required, which was challenging for the inside hypobaric chamber safety attendant who was also simultaneously controlling the airflow rate, and monitoring the $SpO_2$ and blood pressure levels of the participant.

## Methods and materials

### Materials

The protocol described in this peer-reviewed article is published on protocols.io, DOI: dx.doi.org/10.17504/protocols.io.kqdg3xy47g25/v1 and is included for printing as S1 Text with this article.

The protocol described here was utilized for previous studies [1, 2] that followed recommendations for human research by the Simon Fraser University Office of Research Ethics. Written informed consent was given by every participant in accordance with the Declaration of Helsinki. The protocol was approved by the Simon Fraser University Office of Research Ethics.

### Anticipated results

The protocol described here will help in elucidating the extent of differences between NH and HH in terms of electro-neurophysiological, cardiovascular, cerebrovascular and respiratory data. With hypoxia, one can expect an increase in breathing rate, decrease in end tidal carbon dioxide, increase in heart rate, and a decrease in $SpO_2$. Sample responses during 30 minutes (5 minutes eyes-open resting state and 25 minutes of visuospatial attention task) of EEG data, blood pressure data, end-tidal carbon dioxide and oxygen data, breathing frequency data, MCA blood flow velocity, and $SpO_2$ data during the NN, NH and HH conditions described can be viewed in our previous publications [1, 2]. Although the literature is still divided as to whether there are differences between NH and HH, it is suggested that HH will have a lower $SpO_2$ and decreased minute ventilation [11].

In our analysis of the resting state data, we correlated EEG power and multi-scale entropy with $SpO_2$. We found no conditional differences in correlations the $SpO_2$ during the NH condition had plateaued (participants started the HH condition already desaturated) [2];however, we did find that a difference in correlations exists in desaturating compared to desaturated. In our analysis of the task data [1] we found no difference between conditions in the cardiovascular and respiratory data; however, we found that beta power differed between NH and HH that we hope others will replicate as it suggests a difference does exist between NH and HH.

### Summary

Hypoxia research is a broad field ranging from high altitude studies to hypoxic disease at sea level. This protocol will aid in recording EEG data within a hypobaric chamber, and more specifically aid in the standardization of study protocols comparing NH and HH hypoxia. The current literature is lacking in studies directly comparing NH and HH, and this problem is exacerbated by the lack of standardization of study protocols in HH and NH experiments. Alongside the comparison of NH and HH, our study protocol will aid in the development of future experiments on the impact of hypoxia on neurophysiology alongside cardiorespiratory and cerebrovascular measures. Performing multiple simultaneous physiological recordings within a hypobaric chamber is challenging, this protocol will aid in avoiding common issues and allowing the development of a successful experimental design.

## Supporting information

**S1 Text.**
(PDF)

**S2 Text. Medical history questionnaire.**
(DOCX)

## Acknowledgments

We would like to thank the staff of the Environmental Medicine and Physiology Unit for all their support in collecting data for this study. Specifically, we would like to thank Ben Zander, Meagan Abele, and Matt Arnold.

## Author Contributions

**Conceptualization:** Evan A. Hutcheon, Sherri Ferguson, Victoria E. Claydon, Sam M. Doesburg.

**Data curation:** Victoria E. Claydon.

**Investigation:** Evan A. Hutcheon, Victoria E. Claydon, Sam M. Doesburg.

**Methodology:** Evan A. Hutcheon, Sherri Ferguson, Victoria E. Claydon, Urs Ribary, Sam M. Doesburg.

**Project administration:** Evan A. Hutcheon.

**Resources:** Victoria E. Claydon, Urs Ribary, Sam M. Doesburg.

**Software:** Victoria E. Claydon, Sam M. Doesburg.

**Supervision:** Sherri Ferguson, Victoria E. Claydon, Urs Ribary, Sam M. Doesburg.

**Visualization:** Evan A. Hutcheon, Victoria E. Claydon, Sam M. Doesburg.

**Writing – original draft:** Evan A. Hutcheon, Sherri Ferguson, Victoria E. Claydon, Sam M. Doesburg.

**Writing – review & editing:** Evan A. Hutcheon, Sherri Ferguson, Victoria E. Claydon, Urs Ribary, Sam M. Doesburg.

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
