## [Decision Letter · Decision Letter 0]

10 Apr 2024

PONE-D-23-38539A protocol to simultaneously examine cardiorespiratory, cerebrovascular and neurophysiological responses inside a hypobaric chamberPLOS ONE

Dear Dr. Hutcheon,

Thank you for submitting your manuscript to PLOS ONE. After careful consideration, we feel that it has merit but does not fully meet PLOS ONE’s publication criteria as it currently stands. Therefore, we invite you to submit a revised version of the manuscript that addresses the points raised during the review process.

**ACADEMIC EDITOR: **All issues raised by expert reviewers are required. Please provide a point-by-point revision.

We look forward to receiving your revised manuscript.

Kind regards,

Vincenzo Lionetti, M.D., PhD

Academic Editor

PLOS ONE

3. We note that Figure(s) 1 and 2 in your submission contain copyrighted images. All PLOS content is published under the Creative Commons Attribution License (CC BY 4.0), which means that the manuscript, images, and Supporting Information files will be freely available online, and any third party is permitted to access, download, copy, distribute, and use these materials in any way, even commercially, with proper attribution. For more information, see our copyright guidelines: http://journals.plos.org/plosone/s/licenses-and-copyright.

a. You may seek permission from the original copyright holder of Figure(s) 1 and 2 to publish the content specifically under the CC BY 4.0 license. 

Reviewers' comments:

Reviewer's Responses to Questions

**Comments to the Author**

1. Does the manuscript report a protocol which is of utility to the research community and adds value to the published literature?

Reviewer #1: Yes

Reviewer #2: Yes

2. Has the protocol been described in sufficient detail?

To answer this question, please click the link to protocols.io in the Materials and Methods section of the manuscript (if a link has been provided) or consult the step-by-step protocol in the Supporting Information files.

The step-by-step protocol should contain sufficient detail for another researcher to be able to reproduce all experiments and analyses.

Reviewer #1: Partly

Reviewer #2: Yes

3. Does the protocol describe a validated method?

Reviewer #1: Yes

Reviewer #2: No

4. If the manuscript contains new data, have the authors made this data fully available?

Reviewer #1: N/A

Reviewer #2: N/A

**5. Is the article presented in an intelligible fashion and written in standard English?**

Reviewer #1: Yes

Reviewer #2: Yes

6. Review Comments to the Author

Reviewer #1: Method. If a mask leak occurs, the measured end-tidal gases will not be accurate. How is it planned to measure the inspiratory fraction of gases (oxygen, carbon dioxide) in the mask? I mean, in addition to the fact that the internal hypobaric safety assistant will make sure that the reserve bag of respiratory gases never collapses on itself or becomes too full.

Expertise needed to implement protocol “The need for and qualifications of supervising medical personnel depends on the level of the hypoxic stimulus, at a more severe hypoxic dose we recommend having a physician present”. Comment: As this is a research protocol where there may be unexpected events, it is strongly recommended that the physician with expertise in emergency medicine be present at all times.

References. 26 out of 44 manuscripts (59%) are older than ten years ago. Where possible, it is recommended that the references be updated.

Reviewer #2: Comments on manuscript “A protocol to simultaneously examine cardiorespiratory, cerebrovascular and neurophysiological responses inside a hypobaric chamber” by Evan A Hutcheon.

The study, authored by Dr Hutcheon, is a proposal for a protocol to investigate neuro-physiological, cerebro-vascular and cardio-respiratory responses in a hypobaric chamber.

Based on their previous experimental experience, the authors describe a setup designed to address several complex physio-pathological questions related to high altitude physiology by integrating data derived from EEG, transcranial Doppler ultrasound, blood pressure, oxygen saturation, end-tidal gases and respiratory rate.

Major comment

The main problem with this study is that it does not produce any real results, although the subject of the study is worthy of attention in view of all the considerations made by the authors. However, real and reproducible data are needed to confirm feasibility and to evaluate physiological responses to hypoxia in different conditions.

In addition, the description of the protocol setup has already been published in a dedicated journal, as claimed by the authors.

(https://protocols.io/view/aprotocol-to-simultaneouslyexamine-cardiorespirac46cyzaw).

Minor comments

What about hypobaric normoxia (e.g. mountaineers breathing oxygen blends at high altitude)?

What about simulation of acclimatisation responses? In your protocol, depressurisation is quite fast (a few minutes)?

7. PLOS authors have the option to publish the peer review history of their article (what does this mean?). If published, this will include your full peer review and any attached files.

Reviewer #1: **Yes: **Pasquale Longobardi

Reviewer #2: No

---

## [Author Response · Author response to Decision Letter 0]

15 Aug 2024

We have made the changes to our headings and author list. We also added supplementary document captions at the end of the manuscript, but before the references.

We have updated the statement. Due to legal and ethical restriction, data cannot be made publicly available. Data will be made available upon request. Additional published or public analyses would only be permitted with ethics approval for secondary data access. Requests for access to datasets should be made to the corresponding author.

3. We note that Figure(s) 1 and 2 in your submission contain copyrighted images. All PLOS content is published under the Creative Commons Attribution License (CC BY 4.0), which means that the manuscript, images, and Supporting Information files will be freely available online, and any third party is permitted to access, download, copy, distribute, and use these materials in any way, even commercially, with proper attribution. For more information, see our copyright guidelines: http://journals.plos.org/plosone/s/licenses-and-copyright.

a. You may seek permission from the original copyright holder of Figure(s) 1 and 2 to publish the content specifically under the CC BY 4.0 license. 

We bought a license for BioRender and remade Figure 2 with it. 

Figure 1 was commissioned by a co-author (Sherri Ferguson). I have used modified versions of this figure in the past without issue, and this is the first time I am using the original version. I simply stated that it is revised from another paper, which it is.

We have added captions to our supporting information at the end of the manuscript.

Reviewers' comments:

Reviewer's Responses to Questions

Comments to the Author

1. Does the manuscript report a protocol which is of utility to the research community and adds value to the published literature?

Reviewer #1: Yes

Reviewer #2: Yes

2. Has the protocol been described in sufficient detail?

To answer this question, please click the link to protocols.io in the Materials and Methods section of the manuscript (if a link has been provided) or consult the step-by-step protocol in the Supporting Information files.

The step-by-step protocol should contain sufficient detail for another researcher to be able to reproduce all experiments and analyses.

Reviewer #1: Partly

Reviewer #2: Yes

3. Does the protocol describe a validated method?

Reviewer #1: Yes

Reviewer #2: No

I did reference two previous research articles where we used the protocol to generate data. When we applied for ethics for those studies we did not state that the data would be available for secondary uses. 

4. If the manuscript contains new data, have the authors made this data fully available?

Reviewer #1: N/A

Reviewer #2: N/A

5. Is the article presented in an intelligible fashion and written in standard English?

Reviewer #1: Yes

Reviewer #2: Yes

6. Review Comments to the Author

Reviewer #1: Method. If a mask leak occurs, the measured end-tidal gases will not be accurate. How is it planned to measure the inspiratory fraction of gases (oxygen, carbon dioxide) in the mask? I mean, in addition to the fact that the internal hypobaric safety assistant will make sure that the reserve bag of respiratory gases never collapses on itself or becomes too full.

Expertise needed to implement protocol “The need for and qualifications of supervising medical personnel depends on the level of the hypoxic stimulus, at a more severe hypoxic dose we recommend having a physician present”. Comment: As this is a research protocol where there may be unexpected events, it is strongly recommended that the physician with expertise in emergency medicine be present at all times.

Thank you for this consideration. We have included this in the revised manuscript

References. 26 out of 44 manuscripts (59%) are older than ten years ago. Where possible, it is recommended that the references be updated.

We have added some updated references.

Reviewer #2: Comments on manuscript “A protocol to simultaneously examine cardiorespiratory, cerebrovascular and neurophysiological responses inside a hypobaric chamber” by Evan A Hutcheon.

The study, authored by Dr Hutcheon, is a proposal for a protocol to investigate neuro-physiological, cerebro-vascular and cardio-respiratory responses in a hypobaric chamber.

Based on their previous experimental experience, the authors describe a setup designed to address several complex physio-pathological questions related to high altitude physiology by integrating data derived from EEG, transcranial Doppler ultrasound, blood pressure, oxygen saturation, end-tidal gases and respiratory rate.

Major comment

The main problem with this study is that it does not produce any real results, although the subject of the study is worthy of attention in view of all the considerations made by the authors. However, real and reproducible data are needed to confirm feasibility and to evaluate physiological responses to hypoxia in different conditions.

I have cited my results from two previous studies where we published with this data.

In addition, the description of the protocol setup has already been published in a dedicated journal, as claimed by the authors.

(https://protocols.io/view/aprotocol-to-simultaneouslyexamine-cardiorespirac46cyzaw).

The description I published to protocols.io is a requirement for writing a PloS ONE lab protocol.

Minor comments

What about hypobaric normoxia (e.g. mountaineers breathing oxygen blends at high altitude)?

We did not include this condition as we wanted to explicitly compare normbaric and hypobaric hypoxia, as the literature is sparse as to whether both are equivalent. If they are similar, which has to be validated by more studies, then normobaric hypoxia can be utilized as a surrogate in place of a hypobaric chamber for those studies. 

We did include a short section stating that diverse paradigms, such as hypobaric normoxia are possible with our paradigm.

What about simulation of acclimatisation responses? In your protocol, depressurisation is quite fast (a few minutes)?

In theory, acclimatisation protocols are possible; however, these protocols are more complicated and beyond this papers scope. Individuals could stay in the chamber for several days with a commode toilet and food deliveries through the hatch. I have included a short section on how to potentially adapt the paradigm to reflect this.

7. PLOS authors have the option to publish the peer review history of their article (what does this mean?). If published, this will include your full peer review and any attached files.

Do you want your identity to be public for this peer review? For information about this choice, including consent withdrawal, please see our Privacy Policy.

Reviewer #1: Yes: Pasquale Longobardi

Reviewer #2: No

---

## [Decision Letter · Decision Letter 1]

11 Sep 2024

PONE-D-23-38539R1A protocol to simultaneously examine cardiorespiratory, cerebrovascular and neurophysiological responses inside a hypobaric chamberPLOS ONE

Dear Dr. Hutcheon,

Thank you for submitting your manuscript to PLOS ONE. After careful consideration, we feel that it has merit but does not fully meet PLOS ONE’s publication criteria as it currently stands. Therefore, we invite you to submit a revised version of the manuscript that addresses the points raised during the review process.

**ACADEMIC EDITOR: Some issues should be addressed in order to further strenght background. ==============================**

**We look forward to receiving your revised manuscript.**

**Kind regards,**

**Vincenzo Lionetti, M.D., PhD**

Academic Editor

**PLOS ONE**

**Journal Requirements**:

Reviewers' comments:

**Reviewer's Responses to Questions**

**Comments to the Author**

**1. Does the manuscript report a protocol which is of utility to the research community and adds value to the published literature?**

**Reviewer #1: Yes**

**Reviewer #2: Yes**

**2. Has the protocol been described in sufficient detail?**

To answer this question, please click the link to protocols.io in the Materials and Methods section of the manuscript (if a link has been provided) or consult the step-by-step protocol in the Supporting Information files.

**The step-by-step protocol should contain sufficient detail for another researcher to be able to reproduce all experiments and analyses.**

**Reviewer #1: Yes**

**Reviewer #2: Yes**

**3. Does the protocol describe a validated method?**

**The manuscript must demonstrate that the protocol achieves its intended purpose: either by containing appropriate validation data, or referencing at least one original research article in which the protocol was used to generate data.**

**Reviewer #1: Yes**

**Reviewer #2: Yes**

**4. If the manuscript contains new data, have the authors made this data fully available?**

**The PLOS Data policy requires authors to make all data underlying the findings described in their manuscript fully available without restriction, with rare exception (please refer to the Data Availability Statement in the manuscript PDF file). The data should be provided as part of the manuscript or its supporting information, or deposited to a public repository. For example, in addition to summary statistics, the data points behind means, medians and variance measures should be available. If there are restrictions on publicly sharing data—e.g. participant privacy or use of data from a third party—those must be specified.**

**Reviewer #1: N/A**

**Reviewer #2: N/A**

**5. Is the article presented in an intelligible fashion and written in standard English?**

**PLOS ONE does not copyedit accepted manuscripts, so the language in submitted articles must be clear, correct, and unambiguous. Any typographical or grammatical errors should be corrected at revision, so please highlight any specific errors that need correcting in the box below. **

**Reviewer #1: Yes**

**Reviewer #2: Yes**

**6. Review Comments to the Author**

**Please use the space provided to explain your answers to the questions above. You may also include additional comments for the author, including concerns about dual publication, research ethics, or publication ethics. (Please upload your review as an attachment if it exceeds 20,000 characters)**

**Reviewer #1: The manuscript has been diligently revised. A search for similar manuscripts reports for some not cited in your references (see below, ie. Theunissen dated 2022). It remains doubtful that the proposed search protocol is not up to date with the current state of knowledge. I recommend a thorough meta-analysis of the literature**.

1) Theunissen S, Balestra C, Bolognési S, Borgers G, Vissenaeken D, Obeid G, Germonpré P, Honoré PM, De Bels D. Effects of Acute Hypobaric Hypoxia Exposure on Cardiovascular Function in Unacclimatized Healthy Subjects: A "Rapid Ascent" Hypobaric Chamber Study. Int J Environ Res Public Health. 2022 Apr 28;19(9):5394. doi: 10.3390/ijerph19095394. PMID: 35564787; PMCID: PMC9102089.

**2) Neuhaus C, Hinkelbein J. Cognitive responses to hypobaric hypoxia: implications for aviation training. Psychol Res Behav Manag. 2014 Nov 10;7:297-302. doi: 10.2147/PRBM.S51844. PMID: 25419162; PMCID: PMC4234165**

**Reviewer #2: The authors of the present paper have responded thoroughly to the reviewers' questions, thereby improving their work.**

**7. PLOS authors have the option to publish the peer review history of their article (what does this mean?). If published, this will include your full peer review and any attached files**.

**Do you want your identity to be public for this peer review? For information about this choice, including consent withdrawal, please see our Privacy Policy.**

**Reviewer #1: **Yes: **Pasquale Longobardi MD**

**Reviewer #2: No**

****

**While revising your submission, please upload your figure files to the Preflight Analysis and Conversion Engine (PACE) digital diagnostic tool, https://pacev2.apexcovantage.com/. PACE helps ensure that figures meet PLOS requirements. To use PACE, you must first register as a user. Registration is free. Then, login and navigate to the UPLOAD tab, where you will find detailed instructions on how to use the tool. If you encounter any issues or have any questions when using PACE, please email PLOS at figures@plos.org. Please note that Supporting Information files do not need this step.**

---

## [Author Response · Author response to Decision Letter 1]

18 Sep 2024

6. Response to reviewer comments 

Thank you for recommending those two references. They have been added to the manuscript. We have also included the following papers:

Rosales AM, Shute RJ, Hailes WS, Collins CW, Ruby BC, Slivka DR. Independent effects of acute normobaric hypoxia and hypobaric hypoxia on human physiology. Sci Rep. 2022 Nov 15;12(1):19570. doi: 10.1038/s41598-022-23698-5. PMID: 36379983; PMCID: PMC9666440.

Chroboczek M, Kujach S, Łuszczyk M, Grzywacz T, Soya H, Laskowski R. Acute Normobaric Hypoxia Lowers Executive Functions among Young Men despite Increase of BDNF Concentration. Int J Environ Res Public Health. 2022 Aug 30;19(17):10802. doi: 10.3390/ijerph191710802. PMID: 36078520; PMCID: PMC9518314

Hohenauer E, Freitag L, Costello JT, Williams TB, Küng T, Taube W, Herten M, Clijsen R. The effects of normobaric and hypobaric hypoxia on cognitive performance and physiological responses: A crossover study. PLoS One. 2022 Nov 10;17(11):e0277364. doi: 10.1371/journal.pone.0277364. PMID: 36355846; PMCID: PMC9648783.

Hohenauer E, Taube W, Freitag L, Clijsen R. Sex differences during a cold-stress test in normobaric and hypobaric hypoxia: A randomized controlled crossover study. Front Physiol. 2022 Sep 23;13:998665. doi: 10.3389/fphys.2022.998665. PMID: 36225301; PMCID: PMC9549379.

---

## [Decision Letter · Decision Letter 2]

10 Oct 2024

A protocol to simultaneously examine cardiorespiratory, cerebrovascular and neurophysiological responses inside a hypobaric chamber

PONE-D-23-38539R2

Dear Dr. Hutcheon,

We’re pleased to inform you that your manuscript has been judged scientifically suitable for publication and will be formally accepted for publication once it meets all outstanding technical requirements.

Kind regards,

Vincenzo Lionetti, M.D., PhD

Academic Editor

PLOS ONE

Additional Editor Comments (optional):

Reviewers' comments:

Reviewer's Responses to Questions

**Comments to the Author**

1. Does the manuscript report a protocol which is of utility to the research community and adds value to the published literature?

Reviewer #1: Yes

2. Has the protocol been described in sufficient detail?

To answer this question, please click the link to protocols.io in the Materials and Methods section of the manuscript (if a link has been provided) or consult the step-by-step protocol in the Supporting Information files.

The step-by-step protocol should contain sufficient detail for another researcher to be able to reproduce all experiments and analyses.

Reviewer #1: Yes

3. Does the protocol describe a validated method?

Reviewer #1: Yes

4. If the manuscript contains new data, have the authors made this data fully available?

Reviewer #1: N/A

**5. Is the article presented in an intelligible fashion and written in standard English?**

Reviewer #1: Yes

6. Review Comments to the Author

Reviewer #1: The text has been updated with the latest literature. I suggest following the other authors' advice in the protocol, at least as far as deemed appropriate. I recommend publishing the manuscript.

7. PLOS authors have the option to publish the peer review history of their article (what does this mean?). If published, this will include your full peer review and any attached files.

Reviewer #1: **Yes: **Pasquale Longobardi

---

## [Editor Report · Acceptance letter]

14 Oct 2024

PONE-D-23-38539R2 

PLOS ONE

Dear Dr. Hutcheon, 

I'm pleased to inform you that your manuscript has been deemed suitable for publication in PLOS ONE. Congratulations! Your manuscript is now being handed over to our production team.

Kind regards, 

on behalf of

Prof. Vincenzo Lionetti 

Academic Editor

PLOS ONE